# Integrated Modeling of the Catalytic Aquathermolysis Process to Evaluate the Efficiency in a Porous Medium by the Example of a Carbonate Extra-Viscous Oil Field

Ilgiz F. Minkhanov [1,*], Vladislav V. Chalin [1], Aidar R. Tazeev [1], Alexander V. Bolotov [1], Irek I. Mukhamatdinov [1], Sergey A. Sitnov [1], Alexey V. Vakhin [1], Mikhail A. Varfolomeev [1,*], Sergey I. Kudryashov [2], Igor S. Afanasiev [2], Alexey V. Solovyev [2], Georgiy V. Sansiev [2], Dmitry A. Antonenko [2], Kirill A. Dubrovin [2] and Iaroslav O. Simakov [2]

[1] Institute of Geology and Petroleum Technologies, Kazan (Volga Region) Federal University, 18 Kremlyovskaya Str., 420008 Kazan, Russia
[2] JSC Zarubezhneft, Basmanniy Municipal District Intra-City Area, 9 Armianskiy Pereulok, 101000 Moscow, Russia
* Correspondence: minkhanovi@mail.ru (I.F.M.); vma.ksu@gmail.com (M.A.V.)

**Abstract:** In order to evaluate the efficiency of the catalytic aquathermolysis process, physical modeling was carried out on bituminous sediments of Paleocene–Miocene carbonate rocks, characterized by the presence of open and closed fractures. In this context, three filtration experiments were performed on an unextracted reservoir model with extra-viscous oil (EVO). Prior to the experiments, the mineral composition of the rock was determined by X-ray diffraction analysis (XRD) and the content of organic matter and coking products was determined before and after the experiment by thermogravimetric analysis (TGA) as well as the group composition of oil (SARA) before and after the experiment by nuclear magnetic resonance (NMR), gas composition at the fluid separation line, and oil displacement coefficient (ODC). The results of the conducted experiments show that the efficiency of displacement of extraviscous oil could be significantly increased by the use of a solvent and the combined use of a solvent and a catalyst (+9.3% and +17.1% of the oil displacement coefficient, respectively), which is associated with the processes of oil refining.

**Keywords:** extra-viscous oil; steam-thermal treatment; catalyst; oil displacement coefficient

## 1. Introduction

World experience shows that the most widespread and effective methods for development of extra-viscous oil (EVO) reservoirs are thermal, namely steam-thermal treatment (STT). However, this technology has several drawbacks, which reduce its technical and economic efficiency. The main disadvantages of STT are high cost of steam generation and greenhouse gas emissions during its production, rapid watering of the reservoir, while the produced oil after cooling still has high viscosity and density, which complicates its further treatment and transportation.

In work [1], various variants of EVO development using thermal methods were considered. This included the authors' conducted experiments on unextracted rock, where hot water at 100 °C was used as the working agent. However, due to high oil viscosity (600,000 cP), no oil displacement was observed.

A comparison of the methods of EVO extraction is presented in work [2]. During the work three experiments were carried out: displacement with water, exposure with hot water, and steam. In contrast to the first and second experiments, steam injection demonstrated efficiency, with a displacement rate of 13.4% (no oil displacement was observed in the first and second experiments) [3,4].

To increase the efficiency of extra-viscous oil extraction by steam-treatment methods, it is advisable to use combined steam injection with various reagent additives. Such additives

include solvents and catalysts [5–9]. As it is well known, solvents are an effective additive for the process of increasing EVO production, where it is injected together or sequentially with steam [10–12]. After injection, the solvent condenses with the steam at the interface with the oil-saturated rock and further mixes with the oil, resulting in a reduction in oil viscosity and an increase in oil production rate. There are generally five types of solvent injection applications: LASER (liquid addition to steam for enhancing recovery), SAS (steam-alternating solvent), ES-SAGD (expanding solvent-SAGD), SAP (Solvent-Aided Process), and SESF (Solvent Enhanced Steam Flooding).

In the work [13], the authors described the results of filtration experiments on steam injection under different permeability, low initial oil saturation, and content of clay minerals in the rock. Steam injection without additives was ineffective and no oil displacement was observed. However, when adding a rim of the solvent in experiments with similar conditions, the displacement rate was 61–71%, which confirms the effectiveness of its use in conditions of EVO.

The key point is also the choice of solvent type to maximize oil production efficiency. One of the main factors is reservoir temperature [14,15].

The use of catalysts together with steam injection in in situ oil enrichment provides many advantages, one of them being an increase in the degree of oil recovery [16–18].

Catalysts stimulate hydrogenation, hydrogenolysis, hydrolysis, and cracking reactions, leading to improved physical, chemical, and rheological characteristics of oil. The formation of catalytic metals enhances the ability of the rock mineral skeleton to provide oil conversion already in the reservoir. This not only increases reservoir coverage by reducing the molecular weight of resins and asphaltenes, but also irreversibly reduces the viscosity of the produced oil and its content of hard-to-process components [19–25]. In practice, various types of catalysts are used (mineral, water-soluble, oil-soluble, and dispersed), while dispersed catalysts are the most effective in terms of reducing viscosity [26].

In work [27], studies were conducted to evaluate the effectiveness of aquathermolysis catalysts based on salicylic acid and chloride salts. As a result of the interaction of the catalyst with oil, the viscosity of the latter was reduced by 91.5% due to the breaking of C-C, C-N and C-S bonds, and the results of TGA and DSC (differential scanning calorimetry) showed a significant decrease in the proportion of macromolecules of heavy oil.

In work [13] the efficiency of steam injection with solvent and catalyst compared to standard steam injection was considered. The experiments demonstrate an increase in the displacement coefficient by at least 1.5 times.

Laboratory data from previous studies were confirmed by the results of an industrial experiment [28]. For a nickel-based catalytic composition, a reduction in the content of heavy components—resins and asphaltenes—is achieved under reservoir conditions. Now, the most important task is to optimize the cost of the catalytic composition by replacing expensive nickel with iron in its composition. To this end, there is an increasing need to study the effectiveness of catalytic compositions with different ratios of nickel and iron and reduce the cost of the resulting composition without losing the efficiency that was when using a monometallic precursor of a nickel-based catalyst.

The purpose of this work is to study the effect of nanodisperse iron and nickel sulfides on the transformation of the composition of high-viscosity oil in the presence of rock-forming minerals in hydrothermal conditions by determining the displacement coefficient of oil, its properties, and composition after exposure to steam.

## 2. Results and Discussion

Results of filtration experiment № 1. Steam injection at a temperature of 300 °C.

The dynamics of changes in pressure drop and oil displacement coefficient in the first experiment are shown in Figure 1.

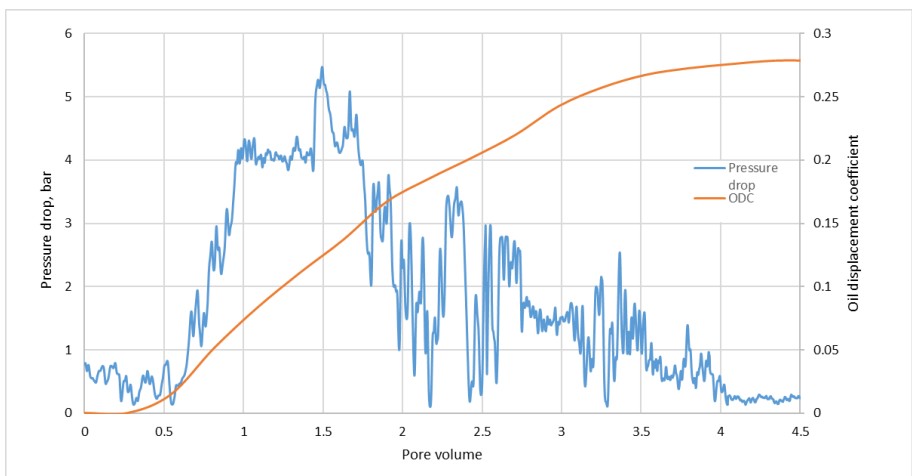

**Figure 1.** Pressure drop and oil displacement coefficient in experiment № 1.

The maximum value of pressure drop is 5.3 bar. The oil displacement coefficient reaches its maximum value at pumping 4.5 pore volume (PV) of steam, and the greatest growth of the displacement coefficient falls on the first 3 PV. The mass of displaced oil during the experiment was 41.9 g, and the displacement coefficient is 27.9%.

Results of filtration experiment № 2. Steam injection with solvent at 300 °C.

The dynamics of changes in pressure drop and oil displacement coefficient in the second experiment are shown in Figure 2.

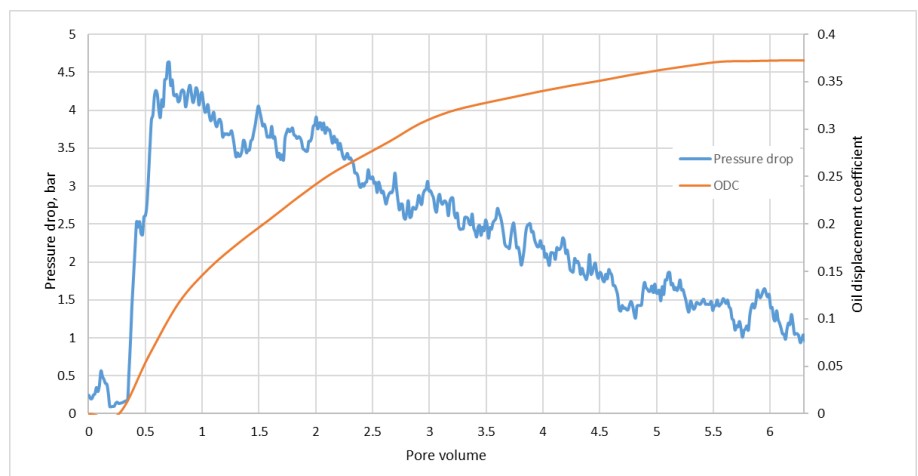

**Figure 2.** Pressure drop and oil displacement coefficient in experiment № 2.

After pumping 0.5 PV, there was a sharp increase in pressure drop to 6 bar. With further injection of steam, the pressure drop in the model decreases to 1 bar. The largest oil displacement corresponds to the interval 0.3–3.0 PV. During the experiment, 55.9 g of oil was displaced, the displacement coefficient of 37.2% was achieved.

Results of filtration experiment № 3. Steam injection with solvent and catalyst addition at 300 °C.

The dynamics of changes in pressure drop and oil displacement coefficient in the third experiment are shown in Figure 3.

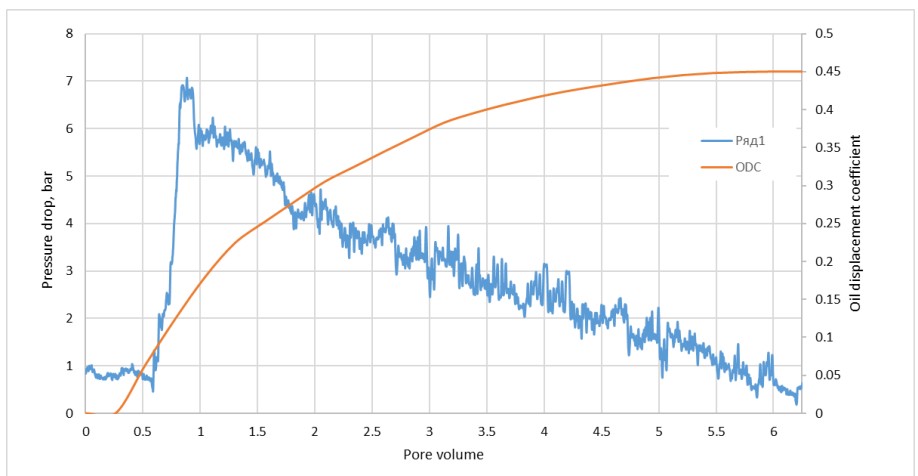

**Figure 3.** Pressure drop and oil displacement coefficient in experiment № 3.

In the experiment, after pumping 0.5 PV there was an increase in depression to 7 bars. After pumping 1 PV of steam, the depression in the model steadily decreased to 0.5 bar. During the experiment 67.6 g of oil was displaced, and oil displacement coefficient is 45.0%.

From the results of physical modeling of oil displacement coefficient determination, in the first experiment with steam injection, (T = 300 °C) oil displacement was 27.9%. The presence of a solvent increases oil displacement by 9.3% and up to 37.2%. In experiment 3, when a catalyst is added, oil displacement equaled 45%, which testifies to the increase in oil displacement efficiency by steam when using a solvent and a catalyst solution, given the increase in oil recovery by 1.3 and 1.6 times, respectively. Summary results of oil displacement coefficients are presented in Figure 4.

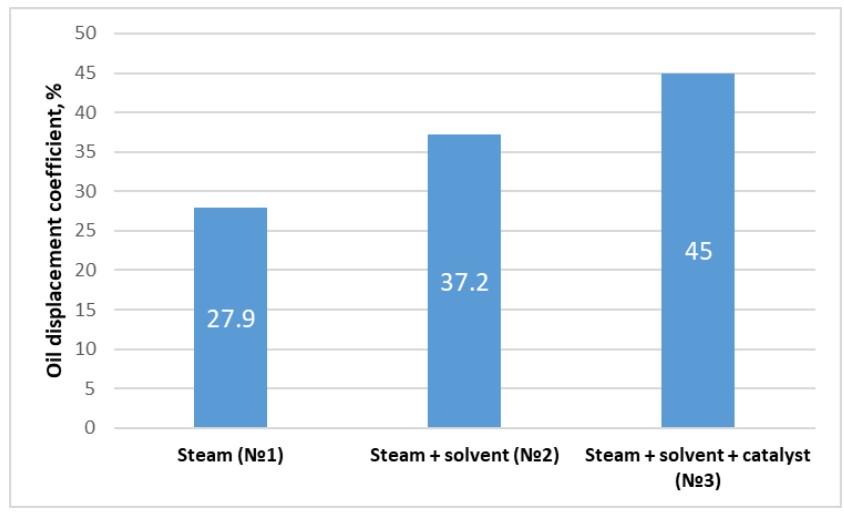

**Figure 4.** Distribution of oil displacement coefficients by experiments.

From the results of analysis of the group composition of displaced oil obtained by NMR, after exposure to steam with a catalyst, the content of asphaltenes in the displaced oil decreases by more than two times compared with steam injection and steam with a solvent, which indicates a thermal transformation of oil in the presence of a catalyst [29]. The results are presented in Table 1.

**Table 1.** Changes in the group composition of oil.

| Sample | Asphaltenes, % | Resins, % | Saturated and Aromatic HC, % |
|---|---|---|---|
| Initial oil | 23.60 ± 0.02 | 33.90 ± 0.07 | 42.50 ± 0.03 |
| Displaced oil 1 | 26.13 ± 0.03 | 34.86 ± 0.02 | 39.01 ± 0.02 |
| Displaced oil 2 | 23.55 ± 0.03 | 43.70 ± 0.02 | 32.75 ± 0.05 |
| Displaced oil 3 | 12.69 ± 0.05 | 43.17 ± 0.04 | 44.14 ± 0.04 |

During the experiments, a wide range of data were obtained after thermogravimetric analysis, namely:

TG—thermogravimetric curve (mass loss curve)—a line characterizing the thermal stability of the sample.

DTG—the first derivative of the mass loss curve—a line characterizing the number of processes under thermal influence.

The results of TGA are shown in Table 2.

**Table 2.** The content of organic matter and coke after experiments.

| Experiment | Content, % | Zone 1 | Zone 2 | Zone 3 | Zone 4 |
|---|---|---|---|---|---|
| 1 | Organic matter | 8 ± 0.02 | 10 ± 0.09 | 12 ± 0.07 | 14 ± 0.01 |
| | Coking products | 0.58 ± 0.03 | 0.59 ± 0.02 | 0.63 ± 0.06 | 1.8 ± 0.03 |
| 2 | Organic matter | 8.28 ± 0.02 | 9.90 ± 0.07 | 12.02 ± 0.08 | 11.25 ± 0.09 |
| | Coking products | 0.65 ± 0.05 | 0.78 ± 0.06 | 0.52 ± 0.02 | 0.88 ± 0.03 |
| 3 | Organic matter | 6.31 ± 0.08 | 6.33 ± 0.04 | 8.95 ± 0.03 | 13.34 ± 0.11 |
| | Coking products | 0.32 ± 0.01 | 0.20 ± 0.10 | 0.58 ± 0.04 | 1.15 ± 0.02 |

According to the results, we observed an increase in organic matter and the amount of coking products from the entrance to the exit of the reservoir model.

Furthermore, in each experiment, the gas phase was sampled at the fluid separation line to determine the component composition. The results are presented in Table 3.

Steam-thermal treatment (STT) at 300 °C significantly affects the increase in the gas-phase content. As can be seen from the data presented, there is a decrease in hydrogen content. Probably, the presence of the catalyst contributes to hydrogenation of double and triple bonds formed during cracking of high-molecular-weight heteroorganic compounds of heavy oil of the Boca de Jaruco field. At a temperature of 300 °C, the presence of iron thallate leads to a decrease in hydrogen sulfide content, as it participates in the formation of the sulfide form after the decomposition of the catalyst precursor [30].

**Table 3.** Component composition of gas at the fluid separation line in the experiments (mol %).

| Component | Experiment № 1 | | | | Experiment № 2 | | | | | | Experiment № 3 | | | | | |
|---|---|---|---|---|---|---|---|---|---|---|---|---|---|---|---|---|
| | Pore Volume | | | | | | | | | | | | | | | |
| | 1 | 2 | 3 | 4 | 1 | 2 | 3 | 4 | 5 | 6 | 1 | 2 | 3 | 4 | 5 | 6 |
| Hydrogen | 14.89 | 12.77 | 13.35 | 12.92 | 2.94 | 3.09 | 4.45 | 3.05 | 3.53 | 4.02 | 5.87 | 5.56 | 5.31 | 5.92 | 5.21 | 6.07 |
| Carbon dioxide | 52.26 | 65.85 | 64.06 | 63.13 | 27.69 | 25.31 | 29.91 | 23.36 | 22.85 | 17.15 | 68.31 | 68.60 | 68.72 | 65.77 | 63.80 | 60.12 |
| Methane | 4.56 | 4.75 | 5.79 | 5.69 | 15.46 | 17.19 | 23.45 | 19.01 | 22.19 | 24.36 | 3.35 | 3.57 | 3.91 | 4.19 | 4.15 | 5.95 |
| Ethan | 1.44 | 1.25 | 1.65 | 1.59 | 3.93 | 4.46 | 4.95 | 5.35 | 6.17 | 7.05 | 0.82 | 0.83 | 0.99 | 1.07 | 1.12 | 1.85 |
| Propane | 1.88 | 0.92 | 1.34 | 1.31 | 3.56 | 3.66 | 3.48 | 4.58 | 5.21 | 6.08 | 12.18 | 13.06 | 13.46 | 13.70 | 16.26 | 14.70 |
| Hydrogen sulfide | 2.04 | 11.08 | 10.89 | 12.34 | 3.14 | 2.92 | 4.02 | 4.01 | 3.91 | 0.13 | 0.67 | 0.66 | 0.84 | 0.90 | 1.01 | 1.77 |
| Iso-Butane | 0.30 | 0.09 | 0.14 | 0.14 | 0.29 | 0.28 | 0.27 | 0.39 | 0.47 | 0.59 | 0.04 | 0.04 | 0.06 | 0.06 | 0.08 | 0.14 |
| Butane | 1.55 | 0.47 | 0.73 | 0.74 | 2.10 | 2.37 | 2.00 | 3.01 | 3.33 | 3.85 | 0.43 | 0.41 | 0.53 | 0.55 | 0.69 | 0.93 |
| Iso-Pentane | 1.73 | 0.19 | 0.15 | 0.17 | 1.20 | 1.07 | 0.78 | 1.11 | 1.10 | 1.37 | 0.09 | 0.09 | 0.11 | 0.13 | 0.14 | 0.26 |
| Pentane | 0.67 | 0.22 | 0.24 | 0.26 | 1.84 | 1.82 | 1.33 | 1.92 | 1.91 | 2.35 | 0.22 | 0.21 | 0.32 | 0.32 | 0.43 | 0.61 |
| Hexane | 17.68 | 2.22 | 1.43 | 1.43 | 24.17 | 25.22 | 16.64 | 22.69 | 19.42 | 23.59 | 0.00 | 0.00 | 0.00 | 0.00 | 0.00 | 0.00 |
| Heptane | 0.58 | 0.11 | 0.14 | 0.16 | 12.45 | 12.03 | 8.44 | 11.18 | 9.59 | 9.47 | 4.27 | 3.93 | 3.12 | 3.52 | 3.61 | 3.59 |
| Octane | 0.37 | 0.07 | 0.10 | 0.14 | 1.21 | 0.57 | 0.28 | 0.34 | 0.34 | 0.00 | 0.14 | 0.15 | 0.06 | 0.09 | 0.11 | 0.54 |
| Nonan | 0.07 | 0.02 | 0.00 | 0.00 | 0.00 | 0.00 | 0.00 | 0.00 | 0.00 | 0.00 | 0.00 | 0.00 | 0.00 | 0.00 | 0.00 | 0.00 |
| CO | 0.00 | 0.00 | 0.00 | 0.00 | 0.00 | 0.00 | 0.00 | 0.00 | 0.00 | 0.00 | 3.61 | 2.90 | 2.58 | 3.79 | 3.39 | 3.46 |
| Amount | 100 | 100 | 100 | 100 | 100 | 100 | 100 | 100 | 100 | 100 | 100 | 100 | 100 | 100 | 100 | 100 |

## 3. Materials and Methods

For the studies a bulk model was prepared from the initial unextracted core, which was ground to a fraction of 0.1 ÷ 1 mm (photo of samples presented in Figure 5) and mineral composition was determined by X-ray analysis (diffractogram of initial sample presented in Figure 6, results presented in Table 4) to identify the presence of clay minerals to assess complicating factors for steam injection 300 °C.

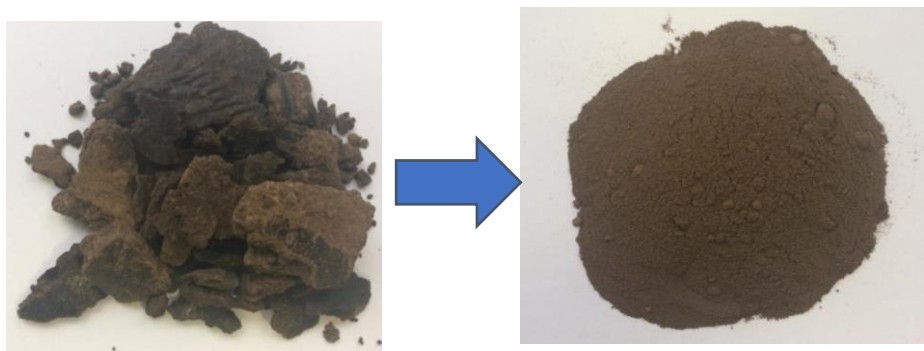

**Figure 5.** Initial and milled to a fraction of 0.1 ÷ 1 mm core material.

**Table 4.** Mineral composition of the rock.

| Mineral | Mass Content, % |
|---|---|
| Calcite | 92 ± 0.08 |
| Pyrite | 1 ± 0.10 |
| Dolomite | 4 ± 0.04 |
| Quartz | 3 ± 0.03 |

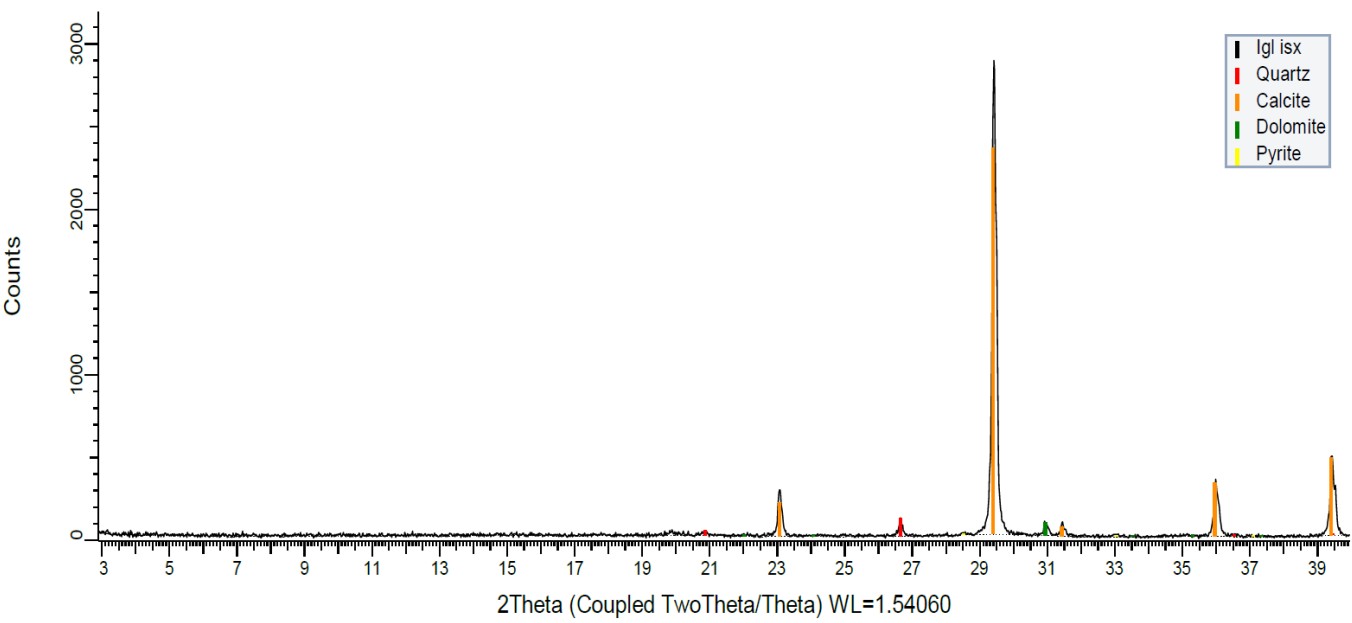

**Figure 6.** Diffractogram of the original core sample.

Based on the XRD results, we can see that the samples are represented by carbonates without clay minerals, which indicates that there is no risk of rock swelling during steam injection.

Initial oil saturation was estimated by two methods: extraction and thermogravimetric analysis (TGA) [31]. The results are shown in Figure 7.

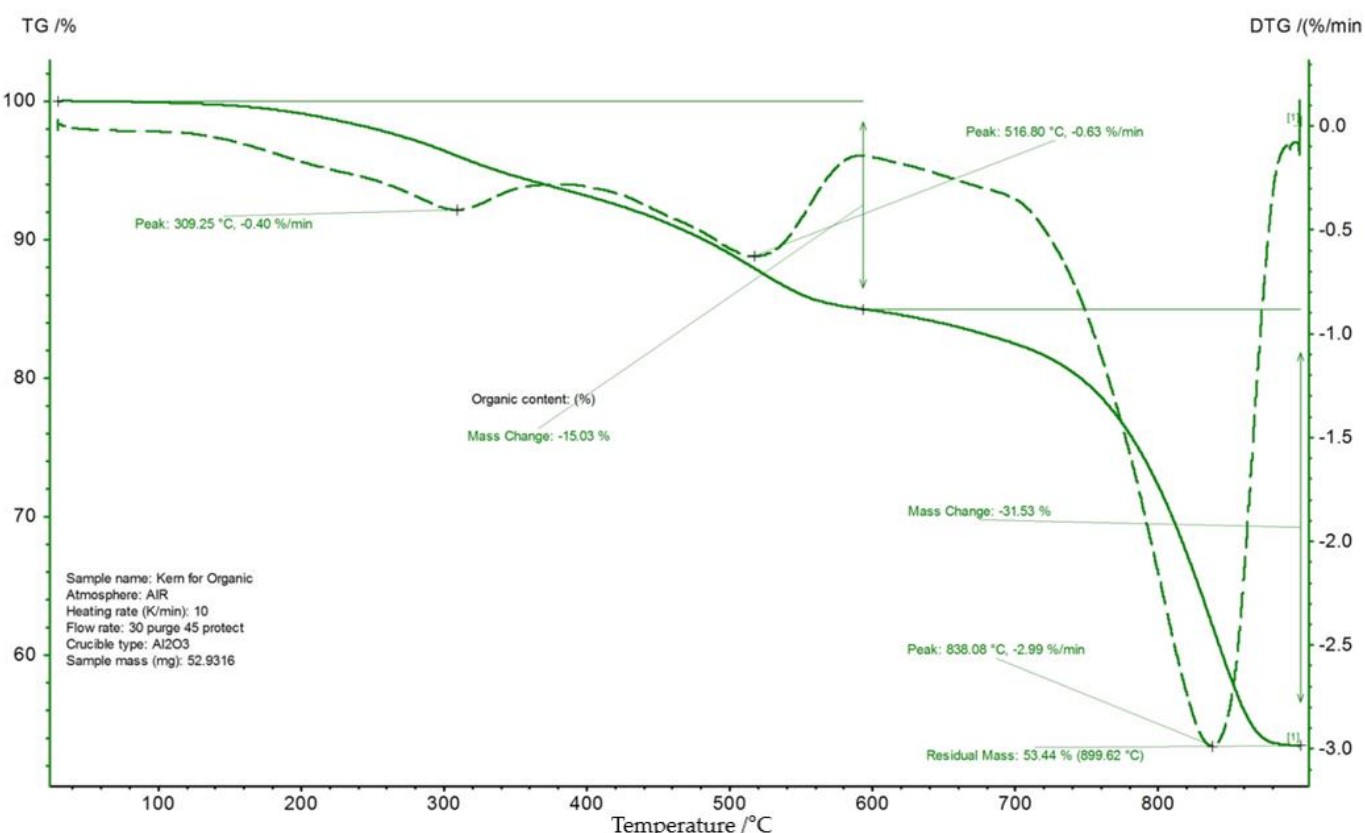

**Figure 7.** Thermogram of the original sample.

During creation of the reservoir model, the milled unextracted rock, in all experiments, was mixed with water of a given mineralization (corresponding to the reservoir, Table 5) in a mass ratio of oil:water equal to 3:1, which corresponds to the initial oil saturation of the reservoir in question; in experiment № 2 to the prepared model was added 3 g of solvent; and in experiment №3 to the prepared model was added 6 g of catalyst solution.

**Table 5.** Component composition of produced water.

| CO$_3^{2-}$ | HCO$_3^-$ | SO$_4^{2-}$ | Cl$^-$ | Ca$^{2+}$ | Mg$^{2+}$ | Na$^+$ + K$^+$ | Total Mineralization | Density, kg/m$^3$ |
|---|---|---|---|---|---|---|---|---|
| 0 | 292.85 | 1119.47 | 7780.54 | 436.51 | 9.63 | 5174.21 | 14,913.21 | 1007.58 |

In this work, a catalyst based on nickel and iron thallates was used [32]. Initially a catalyst precursor was synthesized to study its effect on oil during hydrothermal exposure. At the first stage of the catalyst manufacture, there is a synthesis of sodium salt of fatty acid by interaction of distilled tallow oil with alkali. The fatty acid saponification process can be described by the equation (using oleic acid as an example):

$$C_{17}H_{33}COONa + NaOH \rightarrow C_{17}H_{33}COONa + H_2O \tag{1}$$

The sodium salt of the fatty acid interacts with nickel and iron sulfate (NiSO$_4$, FeSO$_4$) when heated:

$$2C_{17}H_{33}CoNa + NiSO_4 \rightarrow (C_{17}H_{33}COO)_2Ni + Na_2SO_4 \tag{2}$$

$$2C_{17}H_{33}CoNa + FeSO_4 \rightarrow (C_{17}H_{33}COO)_2Fe + Na_2SO_4 \tag{3}$$

As a hydrogen donor, we chose nephras C4-155/205, which is a mixture of naphthenic and aromatic hydrocarbons. It is both a good diluent (dissolves in itself polar and nonpolar components of oil), and can also play the role of hydrogen donor, which during cracking, stops the growth of free radicals and prevents their recombination.

Model composition and experimental conditions are presented in Tables 6 and 7. The pore volume of the model was defined as the difference between the volume of the core holder and the volumes of oil-saturated rock, reservoir water, solvent, and catalyst. Formation water is necessary to create a residual water saturation in the model corresponding to reservoir conditions. Reservoir pressure was created with nitrogen, then the model was heated to a temperature of 300 °C and kept for 24 h.

**Table 6.** Composition of the model of experiments.

| № | Injectant | M$_{rock}$, g | S$_{OI}$, % * | S$_{OI}$, % ** | m$_{oil}$ in the Model, g * | Mass of Formation Water in the Model, g | Mass of Solvent, Catalyst + Solvent, g |
|---|---|---|---|---|---|---|---|
| 1 | Steam | | | | | | - |
| 2 | Steam/solvent | 1000 | 15 | 15.03 | 150 | 50 | 3 |
| 3 | Steam/solvent + catalyst | | | | | | 6 |

\* Initial oil saturation by extraction method, ** Initial oil saturation by TGA method.

The studies were carried out on a unique scientific facility for physical and chemical modeling of in situ combustion and vapor-gravity drainage (Registration Number 2083849, Russian Federation), presented in Figure 8. A detailed description of the setup is presented in [33,34].

**Table 7.** Conditions of experiments.

| № | Temperature, °C | Porosity, % | Pore Volume, mL | Permeability, D | Reservoir Pressure, MPa | Water Injection Rate, mL/min |
|---|---|---|---|---|---|---|
| 1 | | 37 | 187.2 | | | |
| 2 | 300 | 37.5 | 189.7 | 1.6 | 9 | 1.85 |
| 3 | | 37.9 | 191.7 | | | |

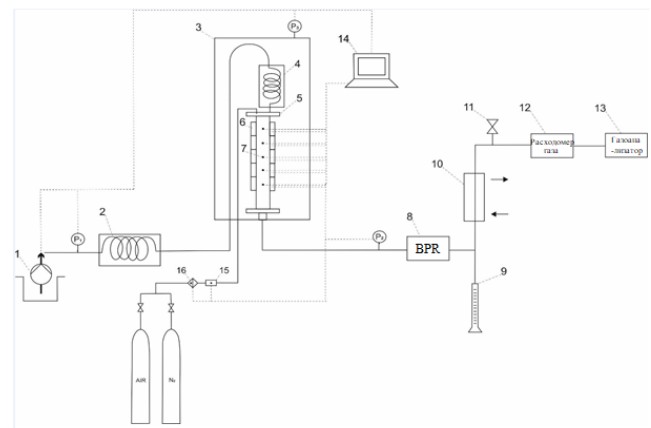 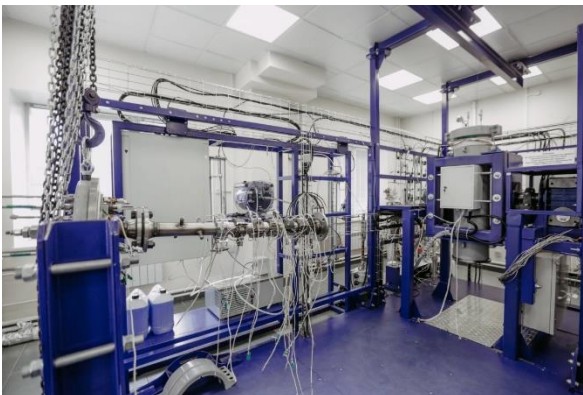

**Figure 8.** Schematic and photo of a unique scientific facility for physical and chemical modeling of the process of in-situ combustion and steam-gravity drainage.

1—high pressure plunger pump; 2—external steam generator; 3—high pressure chamber; 4—internal steam generator; 5—core holder; 6—ceramic electric heaters; 7—thermocouples; 8—back pressure regulator; 9—separating burette; 10—refrigerator; 11—gas extraction line; 12,16—gas flowmeter; 13—gas analyzer; 14—PC; 15—gas flow regulator.

The oil displacement coefficient is calculated from the material balance using the formula:

$$ODC = \frac{m_{e.o.}}{m_{i.o.}} \cdot 100\% \tag{4}$$

where $m_{e.o.}$—mass of extracted oil, $m_{i.o.}$—mass of initial oil.

The model was removed from the core sample holder for further analysis of the model distribution: organic matter content by TGA and group composition of oil by NMR on a Proton 20M NMR analyzer manufactured by CJSC special design bureau Chromatek. The method of determining the group composition of oil is described in [35].

Experiments to determine the thermal characteristics under atmospheric conditions were performed on a TG209 F1 Libra precision thermogravimeter (Netzsch GmbH) combined with an Alpha FTIR spectrometer (Bruker GmbH) in mass signal registration mode according to ASTM E2105–00 (or GOST 57988–2017). Experiments were performed in a dynamic air/nitrogen environment at a linear heating rate of 10 °C/min to 900 °C in 85 µL corundum crucibles.

The composition of inorganic and hydrocarbon gases was determined on a Chromatec Crystal 5000 chromatograph with a flame ionization detector and three thermal conductivity detectors. The sample gas was injected and distributed on three NaX 2 m, NaX 3 m, Hayesep R 3 m, and one DB-1 capillary column (Agilent J&W GC column). The flow rate of the carrier gas (helium) was 15 mL/min. The temperature of the capillary injectors is 200 °C. Temperature of capillary injector—250 °C. Temperature program of the thermostat is 5 min at 60 °C, increasing temperature to 200 °C at a rate of 10 °C/min and holding for 10 min. Chromatec Analytical 3.1 software was used to process the results.

## 4. Conclusions

In the present work, studies on physicochemical modeling of steam-thermal treatment of bituminous oils were carried out. Three experiments were carried out at 300 °C: steam injection, steam injection with solvent, and steam injection with solvent and catalyst. Application of the latter was aimed at additional enrichment of heavy oil, which should increase the final oil recovery. When exposed to steam at high temperatures, various transformations occur in the oil composition, such as steam reforming, formation of carbon monoxide and its further conversion into carbon dioxide and hydrogenation, methanation reactions, which finally lead to production of transformed and lightened oil with lower viscosity.

Experimental results showed that the highest oil displacement was achieved in experiment № 3 (45%), which involved a combination of solvent and steam exposure. In addition, the effect of steam and steam with the solvent on displacement coefficients was 27.9 and 37.2%, respectively. As a result of the study, a deeper conversion of resinous–asphaltene compounds of oil due to their destruction and separation of alkyl substituents is established, resulting in an increase in the content of the fraction of saturated hydrocarbons. Additionally, the SARA analysis of the displaced oil in experiment № 3 confirms an increase in the fraction of saturated and aromatic hydrocarbons and a decrease in the asphaltenes fraction by over 10.9%, compared to the initial oil. This demonstrates the effectiveness of using steam and aquathermolysis catalyst solution in combination in terms of extracting extra-viscous oil.

**Author Contributions:** Conceptualization, I.F.M.; data curation, V.V.C., S.I.K. and I.S.A.; investigation, A.R.T., V.V.C., A.V.S., I.I.M. and S.A.S., G.V.S.; methodology, I.F.M., A.V.B. and D.A.A.; project administration, M.A.V., K.A.D.; supervision, A.V.V.; validation, S.A.S. and I.O.S.; visualization, I.I.M.; writing—original draft, I.F.M.; writing—review and editing, M.A.V. All authors have read and agreed to the published version of the manuscript.

**Funding:** The work is carried out under the support of the Russian Science Foundation related to the Project № 21-73-30023 dated 17.03.2021.

**Conflicts of Interest:** The authors declare no conflict of interest.

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
