# Peer review of "Integrated Modeling of the Catalytic Aquathermolysis Process to Evaluate the Efficiency in a Porous Medium by the Example of a Carbonate Extra-Viscous Oil Field"

_catalysts, doi:10.3390/catal13020283_

Round 1

Reviewer 1 Report

Interesting manuscript on integrated modeling of the catalytic aquathermolysis process. The manuscript evaluates the efficiency in a porous medium by the example of a carbonate extra-viscous oil field. The data provided is interesting, however, it needs to be revised before this manuscript can become suitable for publication. My questions/suggestions are as below:

My questions/suggestions:

1.     Please correct the typos in the manuscript: e.g. the last line in the conclusion, aquatremolysis should be aquathermolysis.

2.     Cross-referencing needs to be corrected. Please check all the figures and table numbers. E.g., the Line above figure 3 says, “the dynamics of changes in pressure drop and oil displacement coefficient in the second experiment are shown in Fig.3”. It should be the No.3 experiment.

3.     Table 1 and figure 5 convey the same data. Keep only one of the items- either a figure or a table. Same thing for table 2 and figure 6.

4.     Table 3 looks cluttered and makes it difficult to understand. Keep significant figures to 2 after the decimal point. (e.g., 3.564 will be 3.56).

5.     In places, the words are confusing. E.g., figure 8 shows XRD results, however, after that the line says, “based on the XRF results”, is it a typo?

6.     Define acronyms, e.g. PV.

Reviewer 2 Report

In this manuscript three oil recovery cases from bituminous sediments of carbonate rocks are comparatively experimented. High oil displacement coefficient is achieved under catalytically steam-sovent exposure conditions. The mineral composition,  the content of organic matter and coking products and the group composition of oil are characterized by means of XRD, XRF, TG, NMR and GC analysis. However, several points are not clear enough as follows.
1 For the experiment conditions for Figure 1, 2, 3, what is pore volume? and how to measure it? What is the time for pressure drop change?
2 The data in Figure 4 seem repetitive with those in Figure 1, 2, 3.
3 The data in Table 1 and Figure 5 seem repetitive. Similarly, the data in Table 2 and Figure 6.
4 In Table 6, what is the solvent, nephras C4-155/205? What is the formation water and its mass?
5 In Table 7, the presssure by what?, the flow rate for what?
6 For Figure 10, the experiment procedure is not described. For example, how are the rock sample, steam, solvent and catalyst arranged, and what is the experiment time?
7 Why are the catalyst (C17H33COO)2Ni and (C17H33COO)2Fe?
8 If there is optimized parameters for this catalytically steam-sovent treatment system.

Reviewer 3 Report

The manuscript ,,Integrated modeling of the catalytic aquathermolysis process to evaluate the efficiency in a porous medium by the example of a carbonate extra-viscous oil field,, it is necessary to significantly improve in order to even consider further assessment. I recommend huge major revision.

It is necessary to expand the abstract, as well as significantly improve its informative quality.

It is also necessary to expand the introduction and expand it with other important papers that can be relied on.

At the end of the introduction, strongly emphasize the importance and novelty of this work.

Add line numbering in the manuscript for better navigation in the manuscript.

This important work can help you improve your introduction. Should be pointed out in the introduction. https://www.sciencedirect.com/science/article/pii/S1385894722053529

Also this one: https://pubs.acs.org/doi/abs/10.1021/ef100230k

It is necessary to improve the quality of the language.

2. Results and discussion: Describe them better and more extensively.

Standard deviations should be added to all tables.

Round 2

Reviewer 3 Report

The manuscript can be accept.